# National Ambulance Surveillance System: A novel method using coded Australian ambulance clinical records to monitor self-harm and mental health-related morbidity

Dan I. Lubman[1,2]*, Cherie Heilbronn[1,2], Rowan P. Ogeil[1,2], Jessica J. Killian[1,2], Sharon Matthews[1,2], Karen Smith[3,4,5], Emma Bosley[6], Rosemary Carney[7], Kevin McLaughlin[7], Alex Wilson[8], Matthew Eastham[9], Carol Shipp[10], Katrina Witt[1,2], Belinda Lloyd[1,2], Debbie Scott[1,2]

**1** Turning Point, Eastern Health, Richmond, Victoria, Australia, **2** Monash Addiction Research Centre and Eastern Health Clinical School, Monash University, Box Hill, Victoria, Australia, **3** Ambulance Victoria, Doncaster, Victoria, Australia, **4** Department of Paramedicine, Monash University, Frankston, Victoria, Australia, **5** Department of Epidemiology and Preventative Medicine, Monash University, Melbourne, Victoria, Australia, **6** Queensland Ambulance Service, Brisbane, Queensland, Australia, **7** New South Wales Ambulance, Rozelle, New South Wales, Australia, **8** Ambulance Tasmania, Hobart, Tasmania, Australia, **9** St John Ambulance Australia (NT) Inc., Casuarina, Northern Territory, Australia, **10** Australian Capital Territory Ambulance Service, Fairbairn, Australian Capital Territory, Australia

* dan.lubman@monash.edu.au

**Data Availability Statement:** The datasets generated and analysed for the current study are not publicly available due to the need to protect

## Abstract

Self-harm and mental health are inter-related issues that substantially contribute to the global burden of disease. However, measurement of these issues at the population level is problematic. Statistics on suicide can be captured in national cause of death data collected as part of the coroner's review process, however, there is a significant time-lag in the availability of such data, and by definition, these sources do not include non-fatal incidents. Although survey, emergency department, and hospitalisation data present alternative information sources to measure self-harm, such data do not include the richness of information available at the point of incident. This paper describes the mental health and self-harm modules within the National Ambulance Surveillance System (NASS), a unique Australian system for monitoring and mapping mental health and self-harm. Data are sourced from paramedic electronic patient care records provided by Australian state and territory-based ambulance services. A team of specialised research assistants use a purpose-built system to manually scrutinise and code these records. Specific details of each incident are coded, including mental health symptoms and relevant risk indicators, as well as the type, intent, and method of self-harm. NASS provides almost 90 output variables related to self-harm (i.e., type of behaviour, self-injurious intent, and method) and mental health (e.g., mental health symptoms) in the 24 hours preceding each attendance, as well as demographics, temporal and geospatial characteristics, clinical outcomes, co-occurring substance use, and self-reported medical and psychiatric history. NASS provides internationally unique data on self-harm and mental health, with direct implications for translational research, public policy, and clinical practice. This methodology could be

privacy and confidentiality. Ambulance data are provided to Turning Point under strict conditions for the storage, retention and use of the data. The current approval permits storage of the data at one site, Turning Point, with any analysis to be undertaken onsite, no data to be removed, and no dissemination of unit level data. Researchers wishing to undertake additional analyses of the data are invited to contact Turning Point as the data custodians at info@turningpoint.org.au.

**Funding:** This work was supported by funding from the Commonwealth Department of Health, the Department of Health and Human Services (Victoria), Beyond Blue and Movember. The funders had no role in study design, data collection and analysis, decision to publish, or preparation of the manuscript.

**Competing interests:** Prof. Lubman has received speaking honoraria from the following: Astra Zeneca, Camurus, Indivior, Janssen-Cilag, Lundbeck, Servier and Shire, and has participated on Advisory Boards for Indivior and Lundbeck. Prof. Lubman and Dr Scott are investigators on an untied educational grant from Seqirus, utilising data from NASS, but is unrelated to the development of this project. The commercial affiliations do not alter our adherence to PLOS ONE policies on sharing data and materials.

replicated in other countries with universal ambulance service provision to inform health policy and service planning.

## Introduction

Each year suicide claims the lives of more than 800,000 people globally, with numbers increasing each year [1]. In Australia, suicide is the leading cause of death for those aged from 15 to 44 years [2], costing the economy $551 million annually [3]. Despite the Australian government spending over $50 million on suicide prevention over the past decade [4], suicide rates have not declined [5]. Critically, suicide deaths represent only the "tip of the iceberg". For each suicide in Australia, there are 11 hospitalisations for intentional self-harm [6] (defined as deliberate self-injury *regardless* of the degree of suicidal intent [7]), and these presentations are also increasing [8–10]. In response, the World Health Organisation (WHO) has identified that monitoring morbidity-related harms as an indicator of progress towards suicide prevention is imperative [11].

Although it is possible to establish surveillance systems using surveys, methodologies that capture representative populations are expensive, and must be maintained over time to enable the capture of trends and patterns. As such, most surveillance systems use routinely collected administrative data [12]. For example, data on Australian suicide deaths are obtained from coroner's records [13]. However, difficulties in determining suicidal intent, including a lack of guidance on such deliberations [14], contribute to lag times of up to four years and renders these data unsuitable for 'real-time' suicide monitoring [15]. Hospitalisation data (patients admitted for treatment) are an alternative data source, but are likely to miss a substantial number of intentional self-harm events as only those with serious physical or mental health issues are likely to be admitted for further treatment. Emergency Department (ED) data are another source and are often used as an "early warning system" [16] to monitor intentional self-harm related presentations [17–21]. ED data are more inclusive than hospitalisation data, as patients who present to ED but are not admitted to hospital will be included.

WHO member states use the International Statistical Classification of Disease and Related Health Problems: 10th Revision (ICD-10) as an epidemiological tool to classify morbidity and mortality in health datasets. ICD-10 provides codes for injury and poisoning related harms in Chapters XIX Injury, Poisoning and Certain Other Consequences of External Causes and XX External Causes of Morbidity and Mortality [22]. Although ICD was developed to monitor the prevalence of health problems in a consistent manner, for a number of reasons ICD-10 codes do not reliably capture, or distinguish between different types of, intentional self-harm. This leads to an underestimation of self-harm in ED and hospitalisation data. First, ICD-10 codes cannot distinguish suicide attempt from self-injury without suicidal intent [23] meaning research based on ICD-10 can only report one catch-all 'intentional self-harm' variable. Further ICD-10 codes do not capture suicidal ideation. This is problematic as understanding the transition from suicidal ideation to suicide attempt informs effective suicide prediction and prevention activities [24]. Second, an intentional self-harm injury must be clearly documented and medically treated while in ED or hospital (if admitted) for an ICD code to be assigned. Therefore, if a patient either did not disclose the injury was self-inflicted, or was not medically treated for the injury, then intentional self-harm ICD-10 codes may not be recorded [23]. Third, some ED information systems only have capacity to record one code. Reliance on a single code means that the presence of a psychiatric disorder, suicide attempt, self-injury without

suicidal intent or alcohol and/or drug intoxication, could be lost and only the physical injury that required treatment will be recorded (e.g., laceration or fracture). This is a major limitation of self-harm surveillance systems based on ICD-10 codes, and particularly for systems reliant on coded ED data.

Enhanced surveillance programs for self-harm, predominantly using text mining techniques to identify suicidal ideation, suicide attempt and self-injury without suicidal intent, can augment information captured in ICD-10 codes [25]. However, these typically require triage or other clinical staff to complete additional training and coding, and sophisticated IT-methodologies. Time, resourcing, and other constraints may lead to the under-estimation of cases and there may be significant lag time for data availability [16]. Natural Language Processing methodologies based on artificial intelligence (AI), offer significant improvements on text mining [26, 27]. However, success of such AI methodologies are reliant on large and long-term datasets with reliably coded data to give the AI computer algorithms sufficient information to precisely replicate human coding capabilities. Also, given that only around one-third of Australians present to the ED following an episode of self-harm, there are likely many more occurrences in the community that do not result in ED presentation or hospitalisation, and are therefore not captured by either enhanced self-harm or more generalised intentional self-harm surveillance systems [28].

Interrogating routine clinical data recorded by paramedics following an ambulance attendance offers a novel avenue for capturing information related to mental health and self-harm outcomes. Indeed, ambulance attendances for such presentations are common in many jurisdictions including the United Kingdom and Australia [29–31], with paramedics frequently the first, and sometimes only, health professional to respond to acute mental health and self-harm presentations in the community [32]. However, international reviews have revealed that, in the case of self-harm incidents, limited literature have examined these cases in detail [33]. Yet paramedic clinical notes provide a rich source of information, including observations made on scene, the context and pattern of self-harm, as well as co-occurrence with mental health symptomatology and alcohol and other drug use.

The National Ambulance Surveillance System (NASS) is an established and internationally unique multi-jurisdictional surveillance system using coded ambulance clinical records [34]. This paper describes the development and application of the self-harm and mental health modules of NASS, including case ascertainment and classification, recording of additional history and risk indicators, and the multiple outputs available. We also provide examples of its utility in practice, as well as implications for informing policy and service planning.

## Materials and methods

### Project development and data coverage

NASS is a surveillance system for self-harm, mental health, and alcohol and other drug-related harms, created from a well-established monitoring program. The development and methodology of the surveillance system, and its original alcohol and other drug module, have been detailed previously [34–37]. This paper describes the self-harm and mental health modules that were added to the original surveillance system [34], building on existing partnerships with jurisdictional ambulance services from across Australia (ACT Ambulance Service, Ambulance Tasmania, Ambulance Victoria, NSW Ambulance, Queensland Ambulance Service, St Johns Ambulance Northern Territory, St Johns Ambulance Western Australia).

Briefly, NASS covers more than 90% of the Australian population across seven of the eight Australian states and territories (Australian Capital Territory, New South Wales, Northern Territory (from 2016), Queensland, Tasmania, Western Australia (from 2020) and Victoria).

Coded NASS self-harm and mental health data are available from the pilot phase (Sep 2012-Aug 2013), in which the proof of concept and system feasibility were established, and an additional three financial years. From this pilot phase, 12 months of coded data is available from the state of Victoria, and data snapshots of one month per quarter (March, June, September and December) are available for other jurisdictions, except Northern Territory and Western Australia. Self-harm and mental health data for males were subsequently coded as a part of the *Beyond the Emergency* project, which investigated the scale and nature of ambulance attendances for men presenting with acute mental health issues and self-harm [37]. NASS coding and reporting was approved through the Eastern Health Human Research Ethics Committee (HREC), with data provision approved by additional HRECs as required by jurisdictional ambulance services.

## Process overview

Fig 1 presents the five steps of NASS data collection and manual coding, which creates a dataset of coded ambulance attendances related to self-harm and/or mental health symptomatology. An electronic Patient Care Record (*e*PCR) was created for every ambulance attendance. Each *e*PCR included details of patient demographics, attendance location and characteristics,

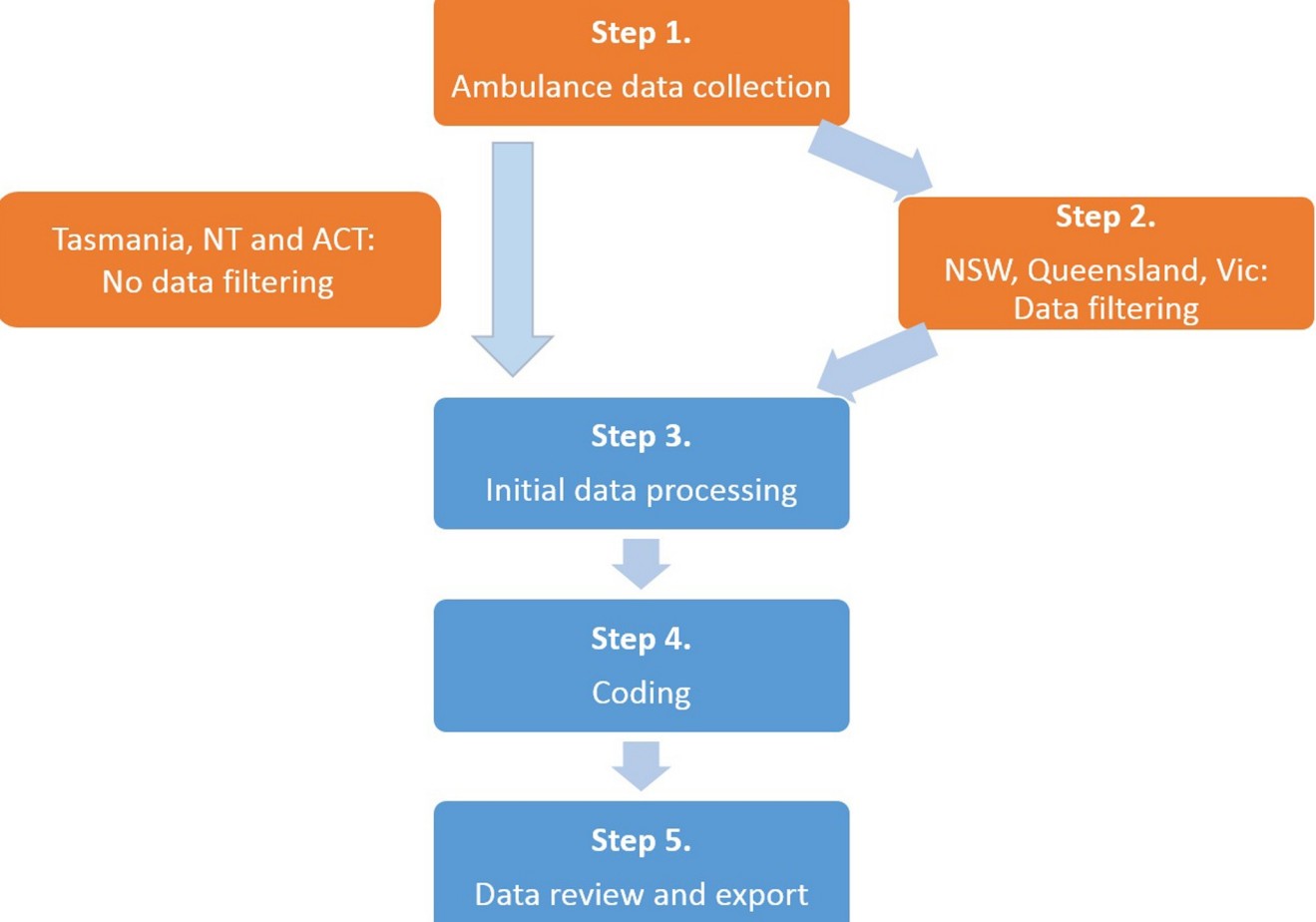

**Fig 1. National Ambulance Surveillance System data collection and coding process.** Processes in orange occur at the jurisdictional ambulance services, with processes in blue occurring at Turning Point.

clinical and treatment information and outcomes and a text description of the paramedic clinical assessment. By using operational ePCRs that were collected for clinical purposes, no additional burden was placed on paramedics. The ePCRs were obtained from electronic clinical information systems used by jurisdictional ambulance services, and were provided to Turning Point, a national addiction treatment and research centre. Case coding involved manual scrutiny of each ePCR by specialist research assistants (RA) to establish (a) case ascertainment and (b) case classification. RAs undergo reiterative training; coding was checked for validity and the project dataset was routinely checked for anomalies. Each step, including detailed description of data filtering, processing, review and exporting, as well as RA training and review, have been previously reported [34]. Coded ambulance data can be reported within three months of an ambulance attendance, depending on data provision and funder requirements.

## Case ascertainment

Inclusion criteria were met if self-harm or mental health symptomatology contributed to the ambulance attendance, using the core inclusion criterion: 'Is it reasonable to attribute a recent (past 24-hours) incident of self-harm or symptom of mental health as contributing to the ambulance attendance?' This information was ascertained through manual scrutiny of each ePCR, considering paramedic clinical assessment, patient self-report, information from third parties and other evidence at the scene, such as written statements of intent (including social media, text messages and written notes), as recorded by paramedics in the ePCR. This evidence was also used to determine method of self-harm as well as presence of risk indicators.

## Case classification

**Self-harm related ambulance attendances.**   Self-harm related ambulance attendances were classified by the presence of self-harm preceding (past 24 hours) or during the ambulance attendance, with four categories of self-harm related ambulance attendances defined and coded as: (a) self-injury (known as non-suicidal self-injury in the US): non-fatal intentional injury without suicidal intent [38]; (b) suicidal ideation: thinking about killing oneself without acting on the thoughts [39]; (c) suicide attempt: non-fatal intentional injury with suicidal intent, regardless of likelihood of lethality [39]; (d) suicide: fatal intentional injury with suicidal intent [39]. Suicide, suicide attempt and suicidal ideation were mutually exclusive, however, self-injury could be simultaneously coded with any other self-harm case category.

**Self-harm method and suicidal ideation preparation.**   Common methods of self-harm were modified from the ICD-10 External Cause Codes [22] and multiple methods could be coded within one case. Thirteen methods of suicide, suicide attempt or suicidal ideation were defined and coded as: (a) intentional alcohol and other drug poisoning (purposeful alcohol and other drug consumption with suicidal intent); (b) carbon monoxide (CO) poisoning; (c) other poisoning (excluding intentional alcohol and other drug poisoning and CO poisoning); (d) hanging; (e) asphyxia (excluding hanging); (f) laceration or penetrating wound; (g) firearm discharge; (h) drowning; (i) jumping from a height; (j) vehicular impact; (k) burn or corrosions; (l) other method; (m) unknown method. Seven methods of self-injury are defined and coded as: (a) laceration or penetrating wound; (b) bodily impact; (c) burn or corrosions; (d) ingestion of foreign object/s; (e) intentional alcohol and other drug poisoning; (f) other method; (g) unknown method. Three categories of suicidal ideation preparation were defined and coded: (a) planned; (b) unplanned; (c) unknown if planned.

**Alcohol and other drug poisoning: Overdose threshold met.**   To compliment the alcohol and other drug module of the NASS [34], a supplementary category classifying the collective impact of substance use in alcohol and other drug poisoning was defined and coded: alcohol

and other drug poisoning (overdose threshold met). This 'overdose threshold met' coding category applies to intentional alcohol and other drug poisoning (defined in this paper), as well as two coding categories described in a previous paper ((a) unintentional alcohol and other drug poisoning: purposeful alcohol and other drug consumption without suicidal intent; (b) undetermined intent alcohol and other drug poisoning: purposeful alcohol and other drug consumption with unknown suicidal intent (where determination of intentional or unintentional alcohol and other drug poisoning cannot be made from the *e*PCR)) [23]. Alcohol and other drug poisoning (overdose threshold met) case inclusion criteria varies depending on the type of drug consumed, using proxy measures to identify cases with potential for medical harm. For alcohol and illicit substances, a potentially life-threatening event was identified by a clinical picture involving a Glasgow Coma Scale (GCS) score of less than nine [40], low respiratory rate and/or paramedic concern for securing an airway. For pharmaceutical preparations, a concordant clinical picture to alcohol or illicit drug overdose, or the consumption of 10 or more times the typically prescribed dose was used to determine alcohol and other drug poisoning. Case inclusion criteria for pharmaceutical drugs varies from that of alcohol and illicit substances due to the complexity of considering total drug effect for individual pharmaceutical preparations during the manual coding process [34].

**Mental health-related ambulance attendances.** Mental health-related ambulance attendances were classified by the presence of a mental health symptom preceding or during the ambulance attendance and, importantly, did not equate to a diagnosis. Four categories of mental health-related ambulance attendances were defined and categorised as: (a) anxiety: overwhelming and intrusive worry, and/or panic attack symptom profile; (b) depression: symptom profile consistent with depression, such as low mood, feelings of hopelessness, despair, worthlessness, anhedonia, change in sleep and/or appetite; (c) psychosis: presence of hallucinations or delusions; (d) other mental health symptom: mental health symptoms not otherwise unspecified. Importantly, cases where presenting mental health symptoms were likely to have a medical cause (e.g., hypoxia, head injury, delirium, diabetes and dementia), rather than a mental health cause, were excluded.

**Mental health and other risk indicators.** For cases ascertained to meet inclusion criteria for mental health-, self-harm- or alcohol and other drug-related ambulance attendances, 41 risk indicators are also coded as 'recorded' or 'not recorded'. Risk indicators were defined and coded into four broad categories, with individual risk factors presented in Table 1: (a) history of self-harm; (b) history of mental health symptoms or diagnosis; (c) concurrent risk indicator: experiencing the risk indicator at the time of ambulance attendance; (d) lifetime risk indicator: have experienced the risk indicator during their lifetime.

## Output variables

The self-harm and mental health modules of NASS capture more than 80 output variables, in addition to the demographic and scene information that was consistent across all modules, including patient details, scene details, and the physical condition of the patient. The self-harm module had 35 variables that categorise the type of self-harm, intent, and method. The mental health module had 46 variables; five that described mental health symptoms at the time of the ambulance attendance, and 31 that described risk indicators. Output variables related to these modules are summarised in Table 1.

## Ethics approval

This project is approved through the Eastern Health Human Research Ethics Committee (HREC), with additional HREC approval for jurisdictional data provision, and requirements

**Table 1. Self-harm and mental health variables available in the NASS dataset.**

| Patient and case details | | | |
| --- | --- | --- | --- |
| **Case details** | **Patient details** | **Scene details** | **Physical condition** |
| *Case number* | Gender | *Public / private* | Fatal event |
| *Case date and time* | Age | *Indoor / outdoor* | *Pulse rate* |
| Transport to hospital | *Residential postcode* | *Event postcode and coordinates* | *Respiratory rate* |
| *Non-transport reason* | | Police co-attendance | *Glasgow Coma Scale* |
| | | Others on scene | *Naloxone administered* |
| | | Minors on scene | Naloxone responsive |
| **Case classification during ambulance attendance** | | | |
| **Mental health** | **Self-harm** | | |
| | **Self-harm and alcohol or other drug poisoning** | **Suicidal intent or planning**\* | **Self-harm method**\*\* |
| Anxiety | Suicide | Not applicable | Intentional alcohol and other drug poisoning |
| | | | Hanging |
| Depression | Suicide attempt | Evidence of intent | Vehicular impact |
| | | Evidence of intent, but denied | Laceration/penetrating wound |
| | | | Jumping from height |
| Psychosis | Suicidal ideation | Suicide plan | Carbon monoxide poisoning |
| | | | Other poisoning |
| | | No suicide plan | Firearm |
| | | | Drowning |
| | | | Burning |
| | | Unknown if plan exists | Asphyxia |
| | | | Other |
| Other/unspecified | Self-injury | Evidence of intent | Intentional alcohol and other drug poisoning |
| | | | Laceration/penetrating wound |
| | | Evidence of intent, but denied | Burning |
| | | | Asphyxia |
| | | | Bodily impact |
| | | | Ingestion of foreign body# |
| | | | Other |
| | Unintentional AOD poisoning | n/a | |
| | Undetermined intent AOD poisoning | n/a | |
| **Relevant history and risk indicators** | | | |
| **Self-harm history** | **Mental health history** | **Current risk indicators** | **Lifetime risk indicators** |
| Suicide attempt | Anxiety | Agitation | Culturally/linguistically diverse |
| Suicidal ideation | Post-traumatic stress disorder | Poor social support | Military service history |
| Self-injury | Obsessive compulsive disorder | Emergency mental health team | Foster care/state guardianship |
| Alcohol and other drug poisoning: unintentional/ undetermined intent | Bipolar disorder | Link to health services | Post-prison release |
| | Depression | Housing problem | Refugee background |
| | Schizophrenia | Unemployment | Suicidal exposure |
| | Other/unspecified psychosis | Bereavement | Intellectual impairment |
| | Borderline personality disorder | Family problem | Acquired brain injury |
| | Other personality disorder | Chronic pain | Dementia |
| | Alcohol and other drug misuse | Sleeping problems | Developmental disorder |

*(Continued)*

**Table 1.** (Continued)

| | | | | |
|---|---|---|---|---|
| | Eating disorder | Financial problems | | |
| | Other / unspecified indicator | Gambling problems | | |
| | | In custody | | |
| | | Bullying | | |
| | | Other / unspecified indicator | | |

* Intent relates to suicidal attempt and self-injury; planning relates to suicidal ideation.

**For suicide, suicide attempt and suicidal ideation, the method pertains to the self-harm method that was undertaken by the patient. For suicidal ideation, the method pertains to the self-harm method that was planned by the patient.

#Excludes alcohol and other drug or other poisons.

Variables that are used directly from ambulance service data provision, and do not undergo additional coding within the NASS, are shown in italics.

for informed consent were waived by these HRECs. Strict protocols are in place for data de-identification, confidentiality, storage, access and reporting. Patient identifiers are provided by some ambulance jurisdictions for the purposes of data linkage. On data receipt, these identifiers are stripped from the dataset and a unique statistical linkage key created. Identifiers are held in a password protected, secure, separate database that is accessible only to database managers. All data is de-identified prior to coding.

## Results and discussion

NASS provides unique and timely monitoring of acute self-harm and mental health morbidity and covers more than 90% of Australia's population. NASS also captures a greater proportion of the community experiencing acute self-harm and mental ill health than surveillance systems that use ED, hospital admission or coroner's data. These highly relevant and valuable data are not captured by other means and strengthen our understanding of the context and burden of self-harm and mental health conditions on individuals, the community, health services, and particularly ambulance responses to acute crises. Further, as mental health symptomatology and suicidal intent fluctuate in both duration and intensity, paramedics' ability to assess these outcomes as close in time as possible to the time of the event are a valuable context for improved public health policy recommendations.

NASS data have already been utilised as part of a major mental health initiative, *Beyond the Emergency*, to inform the development of low-cost mental health interventions that could be provided by ambulance services as well as core mental health training for paramedics [37]. Data have also been used to monitor spatial and temporal trends in self-harm and mental health-related harms [41], and underpinned a range of research to guide public policy, including self-harm and mental health related harms in children and adolescents [42], self-harm and mental health related harms that co-occur with inhalant misuse [43], self-harm and/or mental health in attendances relating to pregabalin misuse [44], co-occurrence of psychosis symptoms in attendances related to methamphetamine use [45], and the role of sleep and other co-morbidities in attendances related to suicide ideation and attempt [46].

System aptitude to delineate types of self-harm and mental health symptomatology, along with method of self-harm and risk indicators, sets it apart from other population level data sources. For example, Fig 2 shows self-harm related ambulance attendances from the three largest jurisdictions (New South Wales, Queensland and Victoria). This highlights NASS's capacity to identify types of behaviours across the spectrum of self-harm, including self-injury, suicidal ideation and suicide attempt. Demarcation of self-harm type facilitates targeted

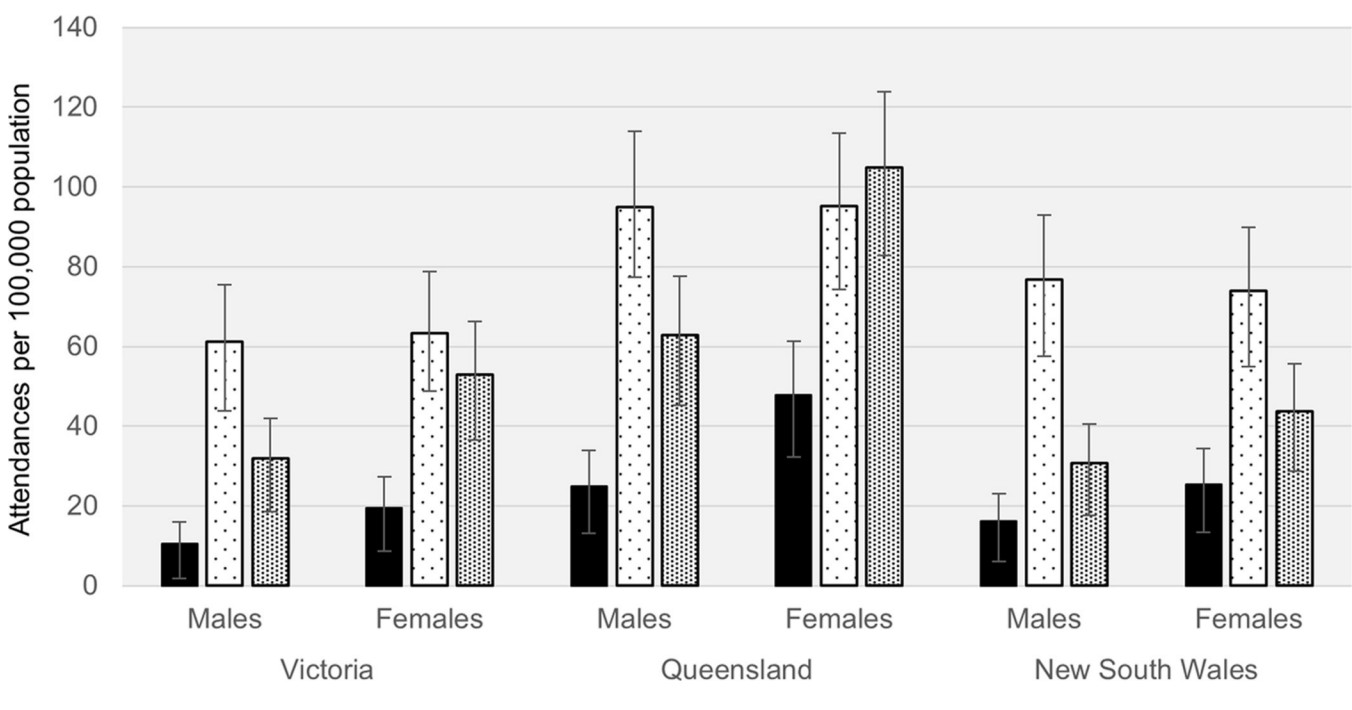

**Fig 2. Types of self-harm related ambulance attendances per 100,000 population in Victoria, Queensland and New South Wales, 2016 snapshot months (Mar, Jun, Sep, Dec).** Error bars represent the binomial proportion confidence interval based on Clopper-Pearson/exact interval.

intervention points and strategies, enhances evaluation of prevention programs, and provides much needed evidence to further clarify the predictive value of previous self-harm on subsequent suicide attempts and fatal suicide [47]. There is a noticeable difference in the rates of specific types of self-harm across the jurisdictions, such as higher rates in Queensland. There are many contributing factors to this, such as treatment access and provision across jurisdictions (e.g., lower access to 24-hour health centres in states such as Queensland that have a higher regional and remote population, and greater geographic spread).

Fig 3 shows the proportion by which specific types of self-harm have co-occurring current mental health symptoms and/or alcohol and other drug use, previous suicide attempts and, for suicidal ideation, whether a current suicide plan is in place. This figure is consistent with previous research highlighting that a high proportion of those who die by suicide in high income countries have mental health diagnoses [48] or substance use disorders [49, 50]. Importantly, our data details patterns of co-morbidity during an acute crisis that can inform the development and evaluation of suicide prevention and early intervention efforts, including identifying at-risk groups. Conversely, as these data can similarly be used to explore self-harm events without co-occurring or historical mental health symptoms.

The type of substances most frequently co-occurring with suicidal ideation and suicide attempt-related ambulance attendances are shown in Fig 4A, while Fig 4B shows data utility by drilling down to individual analgesic medications that were consumed during a self-harm event. These data augment understanding of alcohol and other drug poisoning as an intentional self-harm method, as well as allowing investigation of substance use associated with and preceding self-harm. Because specific pharmaceutical preparations are coded (described elsewhere [34]), rather than broad pharmaceutical groupings, it is possible to make explicit

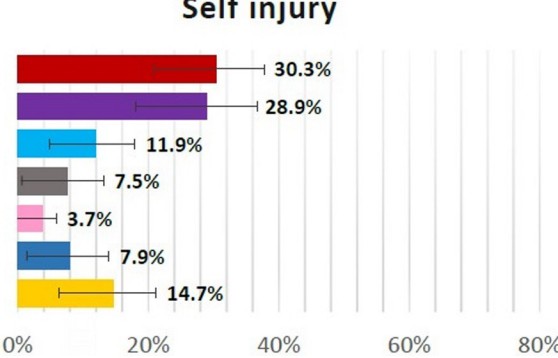

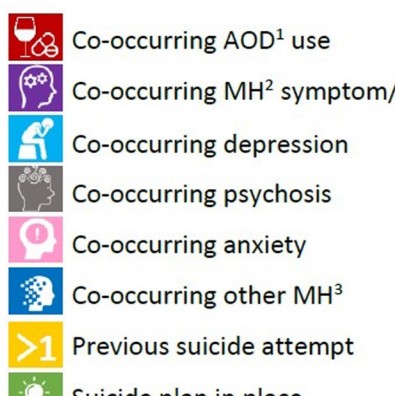

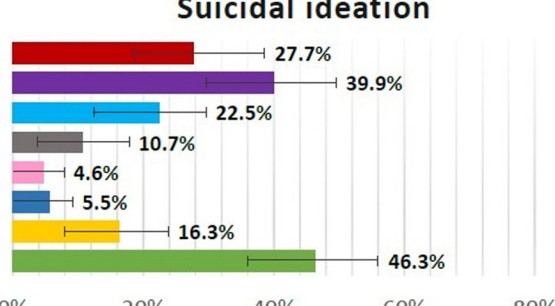

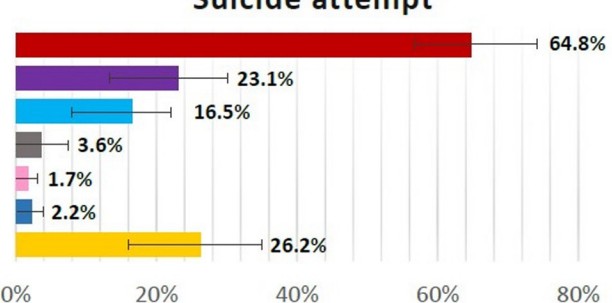

¹ Alcohol or drugs
² Any current mental health symptoms
³ Other mental health symptoms excluding anxiety, depression and psychosis

**Fig 3. Self-harm types by current mental health and alcohol and other drug comorbidity and historical suicide attempts in 2016/2017 financial year.** Error bars represent the binomial proportion confidence interval based on Clopper-Pearson/exact interval.

recommendations for pharmaceutical regulatory schemes and prescribing guidelines to reduce self-harm. For example, these data provide evidence to inform inclusion of specific pharmaceutical medications during the development of real-time prescription monitoring programs [51], such as that recently implemented in the Australian state of Victoria [52]. Capacity to examine acute alcohol and other drug use (including alcohol, specific illicit drugs and individual pharmaceutical medications) immediately prior to self-harm is also critical. Research has historically focused on the well-recognised interactions of alcohol and other drug dependence and self-harm, but the relationship between acute ingestion and self-harm is patchy. Consequently, alcohol and other drug use unrelated to disorders can be overlooked as a clinically relevant issue once the acute intoxication has resolved [53]. Yet, acute alcohol use prior to a suicide attempt has been estimated to increase the risk of suicide attempt by a factor of seven, with an increase to 37 times the risk if alcohol consumption is heavy [54]. Emerging evidence has also shown that acute use of other central nervous system (CNS) depressants, including illicit and pharmaceutical opioids and sedatives/anxiolytics, have almost three times the risk of suicide attempt [55] compared to those who have not used CNS depressants. Our data provides new avenues to examine these associations.

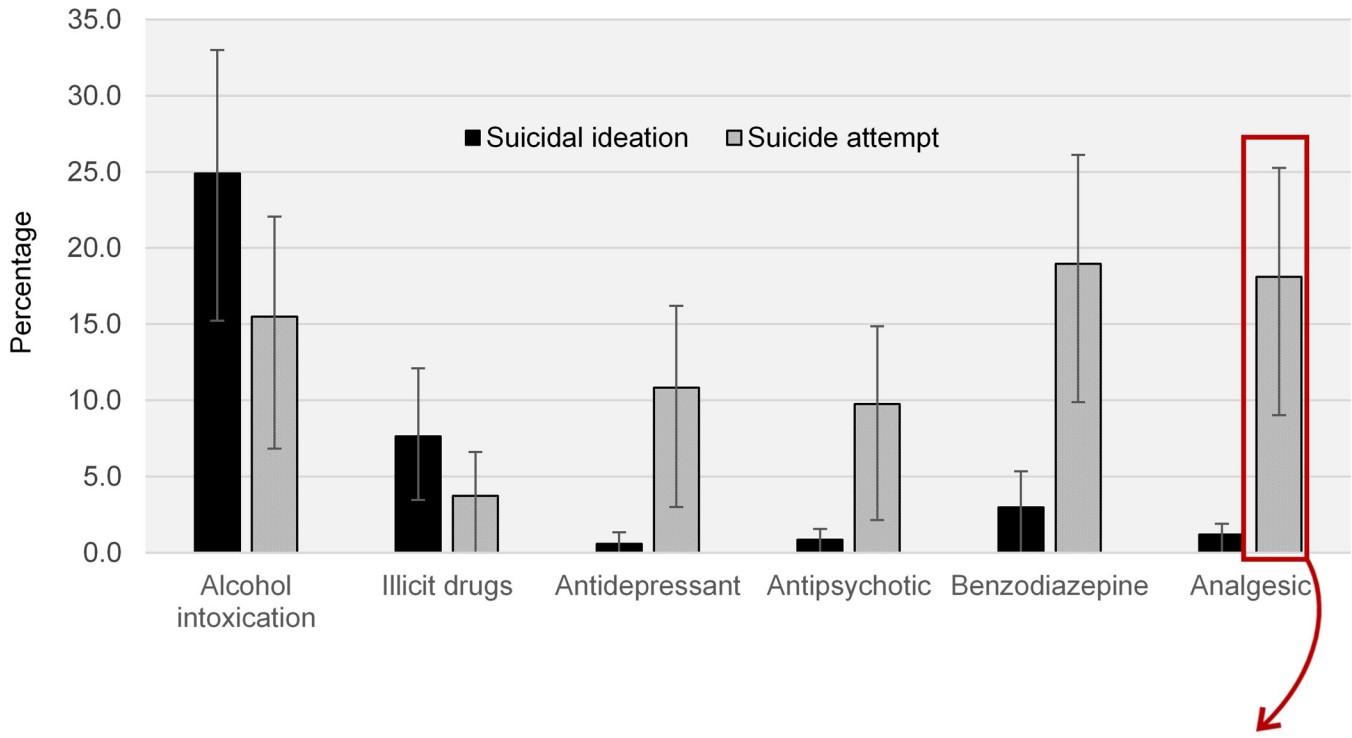

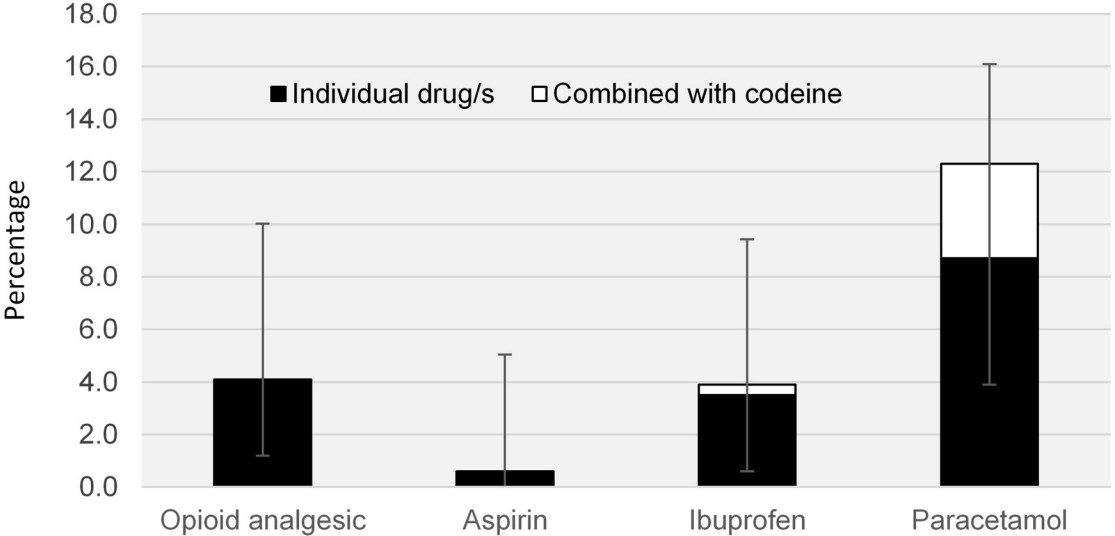

**Fig 4. Drugs involved in Victorian suicide ideation and suicide attempt ambulance attendances.** (A) Percentage of drug categories involved in Victorian suicidal ideation and suicide attempt-related ambulance attendances, 2012 to 2018. (B) Percentage of individual analgesic drugs involved in Victorian suicide attempt-related ambulance attendances, 2012 to 2018. Error bars represent the binomial proportion confidence interval based on Clopper-Pearson/exact interval.

The innovative geospatial mapping capability of this surveillance system NASS was enabled by uptake of geo-coordinate collection by many Australian ambulance services. This technology is associated with significant benefits from a public health perspective, but also carries challenges which should be considered and planned for. On the upside, a key strength is that when combined with other coded health and socio-demographic variables, these geo-spatial coordinates allow for an examination of social aetiologies and health disparities between

different geographical areas [56]. For example, in the context of the present paper, geospatial coordinates may be used to identify local environmental risk factors that are associated with self-harm, for example bridges or rail lines, so that environmental methods can be employed to reduce suicide clusters. An important ethical issue however, is that geospatial co-ordinates create challenges in managing privacy concerns, given that included subjects may be relatively more identifiable. Consequences of a privacy breach are serious and may include family or friends becoming aware of an individual's mental health or self-harm issues, loss of employment (e.g. in cases of illicit drug use), or an inability to obtain health insurance [57]. As such, NASS data are guided by strict protocols for data de-identification, confidentiality, storage, access and reporting (see Ethics section on page 13).

## Limitations and future directions

Data are collected for operational rather than research purposes with paramedics only recording information that they observe or is provided to them by the patient or bystanders, and which they deem clinically relevant to patient care. It is possible that relevant information with respect to self-harm or mental health variables is not recorded, or similar events may not be recorded consistently by different paramedics over time. Close partnerships with jurisdictional ambulance services ensures ongoing, and changing, paramedic and data warehouse operations are clearly understood by the research team. Intra- and inter-rater reliability of the data is maintained by comprehensive training and reliability audits, as outlined in previous publications [34].

NASS data only includes those cases serious enough to require ambulance attendance. Inherently, this dataset is primarily a morbidity dataset. As such, fatal suicide is under-represented as ambulances do not attend all deaths, and when they do attend there may be insufficient information to determine suicidal intent at the scene. However, paramedic clinical records are rich sources of information that complement existing population health metrics (e.g., hospital and ED presentations), and could contribute to linked outcomes that benefit patients. For example, a Scottish study reported that paramedics are well placed to identify people at risk of suicide, having attended a significant proportion of clients in the 12 months prior to them dying by suicide [58]. Hence, analysis of NASS records allow for the identification of numerous risk factors and associated drivers of suicide and self-harm that have potential for intervention either as preventive strategies or treatment options. Importantly, the data are collected in a manner that is not intrusive or demanding on those affected by suicide and self-harm, and does not rely on additional data collection by, or interactions with, already burdened health services.

Transformation of these data into a publicly available, online surveillance resource could be modelled on the ambulance component of *AODstats.org.au*, the online dissemination platform for Victoria's alcohol and other drug state surveillance data [32]. This would enhance self-harm and mental health policy formulation and evaluation at a local, state and national level. The timely nature of the system means data could be available to stakeholders within three to six months of an ambulance attendance, with data uploaded online shortly thereafter. This is significantly more timely than other surveillance systems, including those using coroner's data and enhanced self-harm surveillance systems, with significant time lags [15, 16].

## Conclusions

NASS data provides a population based, cost-efficient resource that can be used to inform the development of prevention initiatives, and serve to evaluate policies and practice over time, as well as specific geographic regions and population groups. NASS is dynamic and changes can

be made in response to emerging or changing mental health harms or priorities. In order to improve the utility of NASS, future work will focus on data linkage and the use of artificial intelligence to assist in screening and coding the clinical data, with the aim of further increasing the timeliness and completeness of the data, as well as its ability to predict future trends and outcomes.

## Acknowledgments

We gratefully acknowledge our project partners in state and territory ambulance services, the paramedics who create the patient care records, and the research and coding team at the National Addiction and Mental Health Surveillance Unit. We would particularly like to thank Foruhar Moayeri, Dhanya Nambiar, James Wilson, Isabelle Hum, and Sam Campbell who assisted with data analysis and/or the figures presented, and Mark Hoffman and Ian Cherrell who maintain the NASS database.

## Author Contributions

**Conceptualization:** Dan I. Lubman, Cherie Heilbronn, Rowan P. Ogeil, Sharon Matthews, Karen Smith, Emma Bosley, Rosemary Carney, Kevin McLaughlin, Alex Wilson, Matthew Eastham, Carol Shipp, Katrina Witt, Belinda Lloyd, Debbie Scott.

**Formal analysis:** Sharon Matthews.

**Funding acquisition:** Dan I. Lubman, Belinda Lloyd, Debbie Scott.

**Investigation:** Dan I. Lubman, Cherie Heilbronn, Rowan P. Ogeil, Jessica J. Killian, Sharon Matthews, Karen Smith, Emma Bosley, Rosemary Carney, Kevin McLaughlin, Alex Wilson, Matthew Eastham, Carol Shipp, Belinda Lloyd, Debbie Scott.

**Methodology:** Dan I. Lubman, Jessica J. Killian, Sharon Matthews, Belinda Lloyd.

**Project administration:** Dan I. Lubman, Debbie Scott.

**Resources:** Dan I. Lubman, Karen Smith, Emma Bosley, Rosemary Carney, Kevin McLaughlin, Alex Wilson, Matthew Eastham, Carol Shipp.

**Supervision:** Dan I. Lubman, Sharon Matthews, Belinda Lloyd, Debbie Scott.

**Visualization:** Cherie Heilbronn.

**Writing – original draft:** Cherie Heilbronn, Rowan P. Ogeil.

**Writing – review & editing:** Dan I. Lubman, Cherie Heilbronn, Rowan P. Ogeil, Jessica J. Killian, Sharon Matthews, Karen Smith, Emma Bosley, Rosemary Carney, Kevin McLaughlin, Alex Wilson, Matthew Eastham, Carol Shipp, Katrina Witt, Belinda Lloyd, Debbie Scott.

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
