## [Decision Letter · Decision Letter 0]

30 Mar 2020

PONE-D-20-00466

National Ambulance Surveillance System: A novel method using coded Australian ambulance clinical records to monitor self-harm and mental health-related morbidity

PLOS ONE

Dear Professor Lubman,

Thank you for submitting your manuscript to PLOS ONE. After careful consideration, we feel that it has merit but does not fully meet PLOS ONE’s publication criteria as it currently stands. Therefore, we invite you to submit a revised version of the manuscript that addresses the points raised during the review process.

We would appreciate receiving your revised manuscript by May 14 2020 11:59PM. To enhance the reproducibility of your results, we recommend that if applicable you deposit your laboratory protocols in protocols.io, where a protocol can be assigned its own identifier (DOI) such that it can be cited independently in the future. For instructions see: http://journals.plos.org/plosone/s/submission-guidelines#loc-laboratory-protocols

We look forward to receiving your revised manuscript.

Kind regards,

Ho Ting Wong, PhD

Academic Editor

PLOS ONE

Additional Editor Comments (if provided):

Please include a paragraph to mention the relation of this manuscript with your previous published paper in PlosOne (i.e. the one you mentioned in the cover letter) and how they are different.

Journal Requirements:

"DL has received speaking honoraria from the following: Astra Zeneca, Camurus, Indivior, Janssen-Cilag, Lundbeck, Servier and Shire, and has participated on Advisory Boards for Indivior and Lundbeck. DL and DS are investigators on an untied educational grant from Seqirus, utilising data from NASS, but is unrelated to the development of this project."

We note that you received funding from a commercial source: Astra Zeneca, Camurus, Indivior, Janssen-Cilag, Lundbeck, Servier and Shire, Seqirus, NASS, Indivior and Lundbeck.

Reviewers' comments:

Reviewer's Responses to Questions

**Comments to the Author**

1. Is the manuscript technically sound, and do the data support the conclusions?

Reviewer #1: Yes

Reviewer #2: Yes

Reviewer #3: Yes

2. Has the statistical analysis been performed appropriately and rigorously? 

Reviewer #1: N/A

Reviewer #2: N/A

Reviewer #3: Yes

3. Have the authors made all data underlying the findings in their manuscript fully available?

Reviewer #1: No

Reviewer #2: Yes

Reviewer #3: No

4. Is the manuscript presented in an intelligible fashion and written in standard English?

Reviewer #1: Yes

Reviewer #2: Yes

Reviewer #3: Yes

5. Review Comments to the Author

Reviewer #1: Review comments for: PONE-D-20-00466

To: The PLOS ONE Editors and Prof. Lubman, et al:

Thank you for the opportunity to review this well-written and clearly argued paper which I recommend for acceptance for publication.

There are a few minor points/recommendations I would like to make:

1) “coronial data” is a term that seems more often used in Australia/New Zealand contexts, and was initially confusing to my American perspective. To reduce initial confusion for global readers, the author team may consider including (coroner) as a parenthetical expression (e.g. coronial (coroner) data) for clarity. If even more clarity was sought, perhaps “national mortality data” may be appropriate as well.

2) The paper helpfully avoids the use of most acronyms except for “AOD”. Like “coronial” above, AOD seems to be used more commonly in Australia/New Zealand. Since there were several places in the text where it is still written out fully, I wonder if it really needs to be shortened and simply writing out “alcohol or other drug” for most of the paper may just be simpler for an international audience?

3) I wish there was more data to discuss, but I will look forward to future follow-up articles that analyze the results of the system described in this paper. After the very well-argued introduction and discussion sections of this paper, I really do wonder how the NASS data compare to coronial data and ED admission data for mental health complaints!

4) There is an aspect of geo-spatial mapping to this project that is just slightly discussed. Given the privacy (downside) and public health (upside) aspects of this technology, I felt a slightly more extensive discussion of this element might have been appropriate.

5) As noted in their disclosure statements, the data is NOT publicly available due to privacy concerns as explained. The authors do state where requests can be sent (Turning Point) if data access was needed.

It was a pleasure to read this well-written and argued paper.

Thank you again for your research efforts.

Reviewer #2: Seems like a reasonable introductory presentation of the data. There are no real statistical methods used in the manuscript. As such there is little for me to evaluate . Suggest the inclusion of error bars in the histograms and appropriate statistical comparisons.

Reviewer #3: This is a well written descriptive account of a novel Australian ambulance surveillance system for coding ambulance records relating to self harm or mental health related morbidity.

PLOS ONE Editorial guidelines state that, "Authors should provide a direct link to the database hosting site from within the paper." This does not yet appear to be publicly available. You state that it is an operational database, not a research database. While valuable, this restricts the utility of the database and limits the value of publishing it in detail.

The results section appears to be a summary of some data held on NASS. I am not convinced that merging of this section with discussion is currently as clear as it could be. Separation of these sections, with a clear description of the aim/questions in the methods section would make the results section more meaningful.

The introduction would have benefited from a global contextual situation of ambulance services and self harm and attempted suicide (See for example Morisson-Rees, S., Whitfield, R., Evans, S., Snooks, H., Huxley, P., John, A., ... & Rees, N. (2015). Investigating the volume of mental health emergency calls in the Welsh Ambulance Service Trust (WAST) and developing a pre-hospital mental health model of care for application and testing. Emerg Med J, 32(5), e3-e3. or Rees, N., Rapport, F., & Snooks, H. (2015). Perceptions of paramedics and emergency staff about the care they provide to people who self-harm: Constructivist metasynthesis of the qualitative literature. Journal of psychosomatic research, 78(6), 529-535.) The discussion would have benefited from a comparison of NASS with other ambulance service data collection systems that enable further analysis - see for example Duncan, E. A., Best, C., Dougall, N., Skar, S., Evans, J., Corfield, A. R., ... & Stark, C. (2019). Epidemiology of emergency ambulance service calls related to mental health problems and self harm: a national record linkage study. Scandinavian journal of trauma, resuscitation and emergency medicine, 27(1), 34.

6. PLOS authors have the option to publish the peer review history of their article (what does this mean?). If published, this will include your full peer review and any attached files.

Reviewer #1: No

Reviewer #2: No

Reviewer #3: No

---

## [Author Response · Author response to Decision Letter 0]

4 Jun 2020

Additional Editor Comments (if provided):

Please include a paragraph to mention the relation of this manuscript with your previous published paper in PlosOne (i.e. the one you mentioned in the cover letter) and how they are different.

We have clarified the differences between our previous paper published in PloS which addressed alcohol and other drug components of NASS, and the present manuscript which examines self-harm and mental health components in the methods section on pages 6-7 (ln 127-) of our revised manuscript. 

Furthermore, on page 9 (ln 181-)of the methods we state the definitions, and variables associated with case ascertainment within the mental health and self-harm modules as follows:

“Self-harm related ambulance attendances were classified by the presence of self-harm preceding (past 24 hours) or during the ambulance attendance, with four categories of self-harm related ambulance attendances defined and coded as: (a) self-injury (known as non-suicidal self-injury in the US): non-fatal intentional injury without suicidal intent (384); (b) suicidal ideation: thinking about killing oneself without acting on the thoughts (395); (c) suicide attempt: non-fatal intentional injury with suicidal intent, regardless of likelihood of lethality (359); (d) suicide: fatal intentional injury with suicidal intent (359). Suicide, suicide attempt and suicidal ideation were mutually exclusive, however, self-injury could be simultaneously coded with any other self-harm case category”. We have removed this from the updated submission files now that the paper has been published, and have updated the reference list accordingly. 

Journal Requirements:

We have updated Fig. 4 presentation to be consistent with PLOS style requirements. This is now also presented in text as: “Fig 4a Percentage of drug categories involved in Victorian suicidal ideation and suicide attempt-related ambulance attendances, 2012 to 2018 and Fig 4b Percentage of individual analgesic drugs involved in Victorian suicide attempt-related ambulance attendances, 2012 to 2018”. 

We have updated the file names for the figures to read “Fig” as requested.

There are ethical restrictions on sharing the data, imposed by the Eastern Health Human Research Ethics Committee (HREC), and the Memoranda of Understandings (MOU) with the jurisdictional ambulance services. These restrictions apply given the sensitive nature of the clinical records described (e.g., they record detailed patient information including unique identifiers such address, geolocation and medical, sociodemographic information pertinent to the ambulance attendance which could be used to identify individuals). In the ethics approval section of the manuscript (page 13) we describe this further:

“This project is approved through the Eastern Health Human Research Ethics Committee (HREC), with additional HREC approval for jurisdictional data provision, and requirements for informed consent were waived by these HRECs. Strict protocols are in place for data de-identification, confidentiality, storage, access and reporting. Patient identifiers are provided by some ambulance jurisdictions for the purposes of data linkage. On data receipt, these identifiers are stripped from the dataset and a unique statistical linkage key created. Identifiers are held in a password protected, secure, separate database that is accessible only to database managers. All data is de-identified prior to coding.”

In our revision, we add further practical examples on the importance of maintaining data integrity and privacy on page 17 in response to comment 4 of reviewer #1.

We recommend the following be added to the data availability statement:

Turning Point has a data request process that is coordinated by Sharon Matthews, who is the Operations Manager for the Population Health Team, Turning Point. Additionally, a secondary, non-author point of contact for data requests is: aodstats@turningpoint.org.au. This email is manned and receives enquires from local council, advocacy groups, health organisations, and members of the general public.

Thank-you for your offer to update the statement on data access and availability based on this further information provided in response to Q2 and Q2a.

"DL has received speaking honoraria from the following: Astra Zeneca, Camurus, Indivior, Janssen-Cilag, Lundbeck, Servier and Shire, and has participated on Advisory Boards for Indivior and Lundbeck. DL and DS are investigators on an untied educational grant from Seqirus, utilising data from NASS, but is unrelated to the development of this project."

We note that you received funding from a commercial source: Astra Zeneca, Camurus, Indivior, Janssen-Cilag, Lundbeck, Servier and Shire, Seqirus, NASS, Indivior and Lundbeck. Please provide an amended Competing Interests Statement that explicitly states this commercial funder, along with any other relevant declarations relating to employment, consultancy, patents, products in development, marketed products, etc.

Here is an amended statement: Prof. Lubman has received speaking honoraria from the following: Astra Zeneca, Camurus, Indivior, Janssen-Cilag, Lundbeck, Servier and Shire, and has participated on Advisory Boards for Indivior and Lundbeck. Prof. Lubman and Dr Scott are investigators on an untied educational grant from Seqirus, utilising data from NASS, but is unrelated to the development of this project. The commercial affiliations do not alter our adherence to PLOS ONE policies on sharing data and materials.

Thank-you for changing the statement online.

Reviewers' comments:

Reviewer #1: Review comments for: PONE-D-20-00466

To: The PLOS ONE Editors and Prof. Lubman, et al:

Thank you for the opportunity to review this well-written and clearly argued paper which I recommend for acceptance for publication.

There are a few minor points/recommendations I would like to make:

1) “coronial data” is a term that seems more often used in Australia/New Zealand contexts, and was initially confusing to my American perspective. To reduce initial confusion for global readers, the author team may consider including (coroner) as a parenthetical expression (e.g. coronial (coroner) data) for clarity. If even more clarity was sought, perhaps “national mortality data” may be appropriate as well.

We thank the reviewer for their positive assessment of the manuscript, and apologise for any geo-specific jurisdictional jargon. We have amended the term coronial as suggested throughout our manuscript.

2) The paper helpfully avoids the use of most acronyms except for “AOD”. Like “coronial” above, AOD seems to be used more commonly in Australia/New Zealand. Since there were several places in the text where it is still written out fully, I wonder if it really needs to be shortened and simply writing out “alcohol or other drug” for most of the paper may just be simpler for an international audience?

We have removed the acronym as suggested in our manuscript revision, and now write alcohol and other drug in full in-text. We have retained the acronym in figure captions and/or tables for brevity, adding a relevant footnote to denote this term.

3) I wish there was more data to discuss, but I will look forward to future follow-up articles that analyze the results of the system described in this paper. After the very well-argued introduction and discussion sections of this paper, I really do wonder how the NASS data compare to coronial data and ED admission data for mental health complaints!

We thank the reviewer for their positive outlook, and look forward to writing further data papers that complement and cite this methods article.

4) There is an aspect of geo-spatial mapping to this project that is just slightly discussed. Given the privacy (downside) and public health (upside) aspects of this technology, I felt a slightly more extensive discussion of this element might have been appropriate.

We agree with the reviewer that this provides both upsides and challenges for researchers and policy makers. To further address these issues, we have modified our Discussion on page 17 which now reads:

“The innovative geo-spatial mapping capability off this surveillance system NASS was enabled by uptake of geo-coordinate collection by some many Australian ambulance services. This technology is associated with significant benefits from a public health perspective, but also carries challenges which should be considered and planned for. On the upside a key strength is that , when combined with the other coded health and socio-demographic variables in these data, these geo-spatial coordinates allow for an examination of social aetiologies and health disparities between different areas (56). For example, in the context of the present paper, geospatial coordinates may be used to identification of local environmental risk factors that are associated with self-harm, for example bridges or rail lines, so that environmental methods can be employed to reduce suicide clusters. An important ethical issue to manage however, is that geo-spatial co-ordinates create challenges in managing privacy concerns, given that individuals may be identifiable. Consequences of a privacy breach are serious and may include: families or friends becoming aware of an individual’s mental health or self-harm issues; loss of employment (e.g., in the case of illicit drug misuse);or an inability to obtain health insurance (57). As such, NASS data are guided by strict protocols for data de-identification, confidentiality, storage, access and reporting (see Ethics section on page 13).”

5) As noted in their disclosure statements, the data is NOT publicly available due to privacy concerns as explained. The authors do state where requests can be sent (Turning Point) if data access was needed.

We have had the following added to the Data availability section of the manuscript:

Turning Point has a data request process that is coordinated by Sharon Matthews, who is the Operations Manager for the Population Health Team, Turning Point. Additionally, a secondary, non-author point of contact for data requests is: aodstats@turningpoint.org.au. This email is manned and receives enquires from local council, advocacy groups, health organisations, and members of the general public.

Reviewer #2: Seems like a reasonable introductory presentation of the data. There are no real statistical methods used in the manuscript. As such there is little for me to evaluate . Suggest the inclusion of error bars in the histograms and appropriate statistical comparisons.

We thank the reviewer for this suggestion and have updated the relevant figures (2-4). The error bars display the binomial proportion confidence interval, based on Clopper-Pearson/exact interval. This information has also been included in the manuscript file under each figure.

Reviewer #3: This is a well written descriptive account of a novel Australian ambulance surveillance system for coding ambulance records relating to self harm or mental health related morbidity.

PLOS ONE Editorial guidelines state that, "Authors should provide a direct link to the database hosting site from within the paper." This does not yet appear to be publicly available. You state that it is an operational database, not a research database. While valuable, this restricts the utility of the database and limits the value of publishing it in detail.

We thank the reviewer for this query, which was also raised by reviewer 1 and the editor. Aggregated data are available to the public via our website (aodstats.org.au), as noted on page 19 of the manuscript. Additionally, the authors collaborate with other researchers and stakeholders on a range or projects which include data from NASS. We have added the following statement to the Data availability section of PloS as requested by the editor:

“Turning Point has a data request process that is coordinated by Sharon Matthews, who is the Operations Manager for the Population Health Team, Turning Point. Additionally, a secondary, non-author point of contact for data requests is: aodstats@turningpoint.org.au. This email is manned and receives enquires from local council, advocacy groups, health organisations, and members of the general public.”

The results section appears to be a summary of some data held on NASS. I am not convinced that merging of this section with discussion is currently as clear as it could be. Separation of these sections, with a clear description of the aim/questions in the methods section would make the results section more meaningful.

The manuscript is a methods paper, and as such its primary goal is to describe the self-harm and mental health symptomatology modules of the National Ambulance Surveillance System (NASS). Given that we don’t present an aim or hypothesis followed by results as per the format of a traditional data paper, we have chosen to present the manuscript in this format. This format is also consistent with previous methods papers published in PloS as noted in our manuscript.

The introduction would have benefited from a global contextual situation of ambulance services and self-harm and attempted suicide (See for example Morisson-Rees, S., Whitfield, R., Evans, S., Snooks, H., Huxley, P., John, A., ... & Rees, N. (2015). Investigating the volume of mental health emergency calls in the Welsh Ambulance Service Trust (WAST) and developing a pre-hospital mental health model of care for application and testing. Emerg Med J, 32(5), e3-e3. or Rees, N., Rapport, F., & Snooks, H. (2015). Perceptions of paramedics and emergency staff about the care they provide to people who self-harm: Constructivist metasynthesis of the qualitative literature. Journal of psychosomatic research, 78(6), 529-535.) The discussion would have benefited from a comparison of NASS with other ambulance service data collection systems that enable further analysis - see for example Duncan, E. A., Best, C., Dougall, N., Skar, S., Evans, J., Corfield, A. R., ... & Stark, C. (2019). Epidemiology of emergency ambulance service calls related to mental health problems and self harm: a national record linkage study. Scandinavian journal of trauma, resuscitation and emergency medicine, 27(1), 34.

We thank the reviewer for providing some internationally relevant resources which we have now incorporated into our revised manuscript in both the introduction and discussion sections. 

In summary, Morrison-Rees and colleagues extracted an electronically generated random sample of 10% of cases in a cross-sectional study and coded the narrative section based upon ICD 10 codes, finding that paramedics often feel ill-equipped for informed clinical decision-making in mental health cases; Rees and colleagues (2015) reviewed the literature and concluded that limited literature has examined paramedic attendances of self-harm events in detail; and Duncan et al. discus opportunities for linkage between ambulance and other hospital departments given they see a high proportion of clients who go on to subsequently die by suicide.

---

## [Decision Letter · Decision Letter 1]

26 Jun 2020

PONE-D-20-00466R1

National Ambulance Surveillance System: A novel method using coded Australian ambulance clinical records to monitor self-harm and mental health-related morbidity

PLOS ONE

Dear Dr. Lubman,

Thank you for submitting your manuscript to PLOS ONE. After careful consideration, we feel that it has merit but does not fully meet PLOS ONE’s publication criteria as it currently stands. Therefore, we invite you to submit a revised version of the manuscript that addresses the points raised during the review process.

We look forward to receiving your revised manuscript.

Kind regards,

Ho Ting Wong, PhD

Academic Editor

PLOS ONE

Reviewers' comments:

Reviewer's Responses to Questions

**Comments to the Author**

1. If the authors have adequately addressed your comments raised in a previous round of review and you feel that this manuscript is now acceptable for publication, you may indicate that here to bypass the “Comments to the Author” section, enter your conflict of interest statement in the “Confidential to Editor” section, and submit your "Accept" recommendation.

Reviewer #1: All comments have been addressed

2. Is the manuscript technically sound, and do the data support the conclusions?

Reviewer #1: Yes

3. Has the statistical analysis been performed appropriately and rigorously? 

Reviewer #1: N/A

4. Have the authors made all data underlying the findings in their manuscript fully available?

Reviewer #1: Yes

5. Is the manuscript presented in an intelligible fashion and written in standard English?

Reviewer #1: Yes

6. Review Comments to the Author

Reviewer #1: Thank you for the opportunity to review your revised paper. Overall, it reads well.

I did notice the new paragraph in the discussion about geo-spatial coordinates had a few typos. Please see attached revision for a few edits.

7. PLOS authors have the option to publish the peer review history of their article (what does this mean?). If published, this will include your full peer review and any attached files.

Reviewer #1: **Yes: **Joseph H. Walline, MD

---

## [Author Response · Author response to Decision Letter 1]

1 Jul 2020

Dear Professor Wong,

Thank-you for the opportunity to revise our manuscript (PONE-D-20-00466R1) entitled “National Ambulance Surveillance System: A novel method using coded Australian ambulance clinical records to monitor self-harm and mental health-related morbidity” for the special issue of PLOS ONE “Digital Health Technology”. We have updated the paragraph on geospatial mapping as suggested by the reviewer. This paragraph now reads:

“The innovative geospatial mapping capability of this surveillance system NASS was enabled by uptake of geo-coordinate collection by many Australian ambulance services. This technology is associated with significant benefits from a public health perspective, but also carries challenges which should be considered and planned for. On the upside, a key strength is that when combined with other coded health and socio-demographic variables, these geo-spatial coordinates allow for an examination of social aetiologies and health disparities between different geographical areas (56). For example, in the context of the present paper, geospatial coordinates may be used to identify local environmental risk factors that are associated with self-harm, for example bridges or rail lines, so that environmental methods can be employed to reduce suicide clusters. An important ethical issue, however, is that geospatial co-ordinates create challenges in managing privacy concerns, given that included subjects may be relatively more identifiable. Consequences of a privacy breach are serious and may include family or friends becoming aware of an individual’s mental health or self-harm issues, loss of employment (e.g. in cases of illicit drug use), or an inability to obtain health insurance (57). As such, NASS data are guided by strict protocols for data de-identification, confidentiality, storage, access and reporting (see Ethics section on page 13).”

Additionally as discussed with Anna Mentsl on 8/6 we have updated the DAS statement in the online system to read:

“The datasets generated and analysed for the current study are not publicly available due to the need to protect privacy and confidentiality. Ambulance data are provided to Turning Point under strict conditions for the storage, retention and use of the data. The current approval permits storage of the data at one site, Turning Point, with any analysis to be undertaken onsite, no data to be removed, and no dissemination of unit level data. Researchers wishing to undertake additional analyses of the data are invited to contact Turning Point as the data custodians at info@turningpoint.org.au."

We hope that you now find our submission suitable for publication.

Kind regards,

Professor Dan Lubman

---

## [Editor Report · Decision Letter 2]

7 Jul 2020

National Ambulance Surveillance System: A novel method using coded Australian ambulance clinical records to monitor self-harm and mental health-related morbidity

PONE-D-20-00466R2

Dear Dr. Lubman,

We’re pleased to inform you that your manuscript has been judged scientifically suitable for publication and will be formally accepted for publication once it meets all outstanding technical requirements.

Kind regards,

Ho Ting Wong, PhD

Academic Editor

PLOS ONE
---

## [Editor Report · Acceptance letter]

21 Jul 2020

PONE-D-20-00466R2 

National Ambulance Surveillance System: A novel method using coded Australian ambulance clinical records to monitor self-harm and mental health-related morbidity 

Dear Dr. Lubman:

I'm pleased to inform you that your manuscript has been deemed suitable for publication in PLOS ONE. Congratulations! Your manuscript is now with our production department. 

Kind regards, 

on behalf of

Dr. Ho Ting Wong 

Academic Editor

PLOS ONE